# Family-Level Multimorbidity among Older Adults in India: Looking through a Syndemic Lens

**DOI:** 10.3390/ijerph19169850

**Published:** 2022-08-10

**Authors:** Sanghamitra Pati, Abhinav Sinha, Shishirendu Ghosal, Sushmita Kerketta, John Tayu Lee, Srikanta Kanungo

**Affiliations:** 1ICMR-Regional Medical Research Centre, Bhubaneswar 751023, India; 2The Nossal Institute for Global Health, Melbourne School of Population and Global Health, The University of Melbourne, Melbourne, VIC 3010, Australia; 3Public Health Policy Evaluation Unit, Department of Primary Care and Public Health, School of Public Health, Imperial College London, London SW7 2AZ, UK

**Keywords:** multimorbidity, family, LASI, India, household, conjugal, siblings, intergenerational

## Abstract

Most evidence on multimorbidity is drawn from an individual level assessment despite the fact that multimorbidity is modulated by shared risk factors prevailing within the household environment. Our study reports the magnitude of family-level multimorbidity, its correlates, and healthcare expenditure among older adults using data from the Longitudinal Ageing Study in India (LASI), wave-1. LASI is a nationwide survey amongst older adults aged ≥45 years conducted in 2017–2018. We included (*n* = 22,526) families defined as two or more members coresiding in the same household. We propose a new term, “family-level multimorbidity”, defined as two or more members of a family having multimorbidity. Multivariable logistic regression was used to assess correlates, expressed as adjusted odds ratios with a 95% confidence interval. Family-level multimorbidity was prevalent among 44.46% families, whereas 41.8% had conjugal multimorbidity. Amongst siblings, 42.86% reported multimorbidity and intergenerational (three generations) was 46.07%. Family-level multimorbidity was predominantly associated with the urban and affluent class. Healthcare expenditure increased with more multimorbid individuals in a family. Our findings depict family-centred interventions that may be considered to mitigate multimorbidity. Future studies should explore family-level multimorbidity to help inform programs and policies in strategising preventive as well as curative services with the family as a unit.

## 1. Introduction

The current demographic trends in low- and middle-income countries (LMICs) project a surge in the ageing population [1]. The rise in the ageing population has led to an increase in chronic conditions among LMICs such as in India, where non-communicable diseases (NCDs) tend to occur around 45 years of age, a decade earlier than in high-income countries (HICs) [2]. This is exacerbated by multimorbidity, the co-occurrence of two or more chronic conditions in an individual [3], which often leads to compromised patient outcomes such as deteriorated physical health; inferior quality of life; and increased healthcare utilisation and expenditure [4,5]. Additionally, multimorbidity has evidently been associated with falls and frailty among adults aged 60 years and above [6,7]. Our recent study based on nationally representative data in India estimated the prevalence of multimorbidity to be around 50% among adults aged ≥45 years [8]. Additionally, inadequately integrated and disease-specific care management programs may accentuate these issues, impeding the provision of appropriate quality of care [9]. Hence, notable health system reforms and policy changes are required to mitigate multimorbidity [10].

In India, living with extended family is common, where around 77% of the older adults reside together with their married adult children [11]. Relatives beyond the nuclear family live and share the same domestic environment and health-promoting factors [12]. They are exposed to and influenced by each other’s lifestyles and activities, such as tobacco/alcohol use and dietary habits [13]. Genetically related coresidents may share predisposing elements that can lead to comparable illness [14]. Previous evidence largely based on inferences drawn from HICs suggest concordance in long-term conditions among conjugal [15,16], sibling [17], and parent–child (intergeneration) pairs [17,18]. Studies suggest spouses may influence each other’s physical and mental health along with health behaviour [19]. Nonetheless, most epidemiological and public health interventions for chronic diseases are directed at the individual rather than the family as a unit, though the latter has immense potential with regard to formulating successful interventions.

A study from India demonstrated substantial concordance in chronic conditions among the family members residing together in the same household [20]. However, the literature is limited by the investigation of risk factor concordance for chronic diseases with no evidence on the prevalence of family or household-level multimorbidity. To reorient the focus of multimorbidity research from the individual to the family as a unit, we propose a new term, “family-level multimorbidity”, defined as two or more members of a family having multimorbidity.

In India, the health system comprises both the publicly funded as well as the private sector. The public healthcare system comprises a three-tier structure, encompassing the primary, secondary, and tertiary levels, which strives to provide equitable and quality-based care. Public healthcare aims to reduce out-of-pocket expenditure by providing services for free or for a minimal user fee. However, the private sector that runs parallel to the public healthcare system is also used by a large number of people and users have to pay for the services. Additionally, the costs of care in the private sector are not regulated, which leads to catastrophic health expenditure. Catastrophic health expenditure remains an additional challenge in seeking continuity of care for multimorbidity. It not only affects individuals but forces families into poverty. A study from Brazil found multimorbidity to be associated with catastrophic health expenditure, especially among the lower strata [21]. Hence, examining multimorbidity at the family level may be critical for developing culturally acceptable and cost-effective family-centred primary healthcare interventions to manage multiple chronic illnesses. Hence, we aimed to estimate the magnitude of family-level multimorbidity, its correlates, and outcomes in terms of healthcare expenditure among older adults using data from the Longitudinal Ageing Study in India (LASI), wave-1. We also estimated the prevalence and correlates of multimorbidity at a conjugal level among sibling and intergenerational pairs.

## 2. Materials and Methods

### 2.1. Overview of the Data

This study is based on the secondary data collected during the first wave of the LASI, the first nationally representative study on the ageing population, conducted throughout India (except Sikkim) by the International Institute for Population Sciences (IIPS), Mumbai in collaboration with the Harvard TH Chan School of Public Health and the University of Southern California from April 2017 to December 2018. A priori criteria of one or more persons residing in a house with a cooking facility and at least one member aged ≥45 years was considered to identify LASI-eligible households (LEHs) following a multistage stratified area probability cluster sampling design. A three-stage sampling was employed in rural areas, whereas a four-stage sampling process was adopted in the urban areas to select the individuals. In each state/union territory, subdistricts or taluks were randomly chosen as the primary sampling unit (PSU) from the sampling frame based on the 2011 census data. In the next stage, village and urban wards were randomly selected from each of the selected subdistricts. Furthermore, a census enumeration block (CEB) was selected from each urban ward. In the last stage, LEHs were randomly selected from each village and CEB to reach the ultimate sampling unit. Complying with the inclusion criteria, 72,250 individuals from 42,949 households formed the ultimate sample size of the LASI.

The LASI conducted community-based face-to-face interviews. An individual survey tool was administered to the household members aged ≥45 years and their spouses (irrespective of age). Another questionnaire recorded data related to household amenities, which was completed by the head of the household. We used four datasets named as “individual”, “household roster”, “household data”, and “biomarker”. The dataset, “household roster” was used to identify the family members and their relationship with each other to produce family-level evidence. The “individual” dataset contained participants’ disease profile, age, education, caste, religion, national region, and expenses due to outpatient visits and hospitalisation. Additionally, the “biomarker” dataset contained individuals’ height and weight, which was used to measure body mass index (BMI). The dataset named “household data” was used to retrieve information on the number of members residing in a household, household income, source of drinking water, type of fuel used for cooking, and if any member smokes inside the house. The LASI reported a non-response rate of 12.7%. The complete description of the methods employed by the LASI is portrayed on their website [22].

In our study, the term “family” refers to two or more members coresiding in the same household. Considering this, 22,526 families became part of this study with members aged ≥45 years. These 22,526 families were further grouped into four subgroups based on the individual respondent’s relationship with the head of the household: conjugal, which includes the household head and his/her spouse; head of the household and his/her brothers or sisters is termed as siblings; and intergenerational, which is the head of the house along with his/her offspring, parents, and grandchildren. Out of the 22,526 families, 19,737 met the above-mentioned criteria for the conjugal group, 353 were grouped in the sibling group, and 2991 formed the intergenerational group (Figure 1). These categories are not exclusive, as some families met the inclusion criteria for more than one category.

### 2.2. Variables

#### 2.2.1. Independent Variables

Based on an extensive review of the previous literature, we included average household age, residence, mean years of education, caste, number of family members, region, and wealth index as independent variables to assess their association with family-level multimorbidity [23,24,25]. We observed these variables to be associated with multimorbidity in various settings. We considered average age at the family level, calculated as the total age of all the eligible family members divided by the number of members included from that family. Similarly, average years of education was calculated for the household. Other covariates, such as residence (urban/rural), caste (Scheduled Caste/Scheduled Tribe/Other Backward Class/none of them), and region of the country (north/east/west/south/central/northeast) were accessed from the individual dataset. Additionally, “household wealth”, which was based on monthly per capita expenditure, source of “cooking fuel”, “drinking water”, “number of members in household” irrespective of age, and “if any member smokes inside the house” were obtained from the household dataset.

#### 2.2.2. Outcome Variables

Multimorbidity defined as two or more chronic conditions in an individual was our primary outcome of interest. We considered a total of seventeen chronic conditions to assess the multimorbidity status, out of which sixteen conditions were self-reported based on the question, “Has any health professional ever diagnosed you with the following chronic conditions or diseases?” with responses of “yes” or “no”. The sixteen long-term conditions were hypertension, diabetes, cancer, chronic lung disease, chronic heart disease, stroke, arthritis and osteoporosis or other bone/joint diseases, neurological or psychiatric problems, hypercholesterolemia, thyroid disease, gastrointestinal problems, chronic renal disease, chronic oral conditions (bleeding and swelling gums, loose teeth, and dental caries/cavity), skin diseases, visual impairment, and hearing impairment, whereas obesity was calculated from weight in kg (measured by the Seca 803 digital weighing machine) and height in m^2^ (measured using a stadiometer). BMI was calculated using the World Health Organization’s (WHO) South Asian cut-off to categorise as obese or not [26].

Out-of-pocket health care expenditures (collected for the year 2017–2018) for the last outpatient department (OPD) visit of all the members of a family were added and divided by the number of family members to find the average family OPD expenditure. Similarly, the last four hospitalisation costs for all the family members were summed together and an average inpatient department (IPD) cost for the family was calculated.

### 2.3. Statistical Analysis

Descriptive statistics for continuous variables (average family age and average years of education) were expressed as mean (±SD) with range. Prevalence was calculated for each categorical variable, presented as frequencies and proportions (*n*, *n*%). Family-level out-of-pocket expenditure on healthcare for both OPD and IPD visits was stated as the median with interquartile range. In the multivariate models, the categorical variables adjusted were average household age, place of living, average years of education of all the members, caste, different regions of India, number of household members, type of fuel used for cooking, if any of the members smoke inside the house or not, source of drinking water, and household wealth index. We used separate multivariable logistic regression models using the “logistic” command to assess correlates for family (two or more members of a family having multimorbidity vs. none or one member), conjugal (none or either partner vs. both partners), sibling (none or one vs. two or more siblings), and intergenerational (none or one generation vs. two or more generations) multimorbidity, expressed as an adjusted odds ratio (AOR) with a 95% confidence interval (CI). A *p*-value < 0.05 was considered as statistically significant. The coefficient of variance (r^2^) was considered to pick the best-fit regression model. This analysis was conducted using STATA, version 17.0 (Stata Corp, College Station, TX, USA).

### 2.4. Ethical Considerations

This study was conducted using anonymous data from the LASI. The LASI obtained ethics clearance from the IIPS, Mumbai and the Indian Council of Medical Research, New Delhi. Prior informed written consent was obtained from all participants.

## 3. Results

The average size of a family was determined to be 5.12 ± 2.52 (mean ± sd) members with a mean family age of 56.81 ± 9.15 years (mean ± sd). Most of the participants were from a rural setup (66.11%), belonged to the other backward class (38.27%), and hailed from the southern part (23.49%) of the country (Table 1).

The prevalence of family-level multimorbidity was 44.6%. In total, 23.32% of the families did not report multimorbidity among any of the members, whereas 40.5% of families with two members had multimorbidity (Appendix A). Amongst the two member families, 33.50% of families reported one member with multimorbidity, whereas in 41.83% families both the members had multimorbidity (Table 2).

Conjugal multimorbidity was prevalent among 41.48% of the households, whereas among 25.18% of the households, none of the spouses had multimorbidity (Table 3). Among the LEHs, household heads and their siblings were coresiding in 353 families, out of which 41.64% of the families reported multimorbidity to be prevalent among two or more siblings. Furthermore, the LASI reported 2991 families with three generations living in the same household, where intergenerational multimorbidity was prevalent among 46.07% of the families.

Multivariable logistic models were executed separately for family-level, conjugal, sibling, and intergenerational multimorbidity. The coefficient of variance r^2^ was 0.129, 0.124, 0.195, and 0.092 for family-level multimorbidity, conjugal, sibling, and intergenerational models, respectively. It demonstrated significantly higher odds of developing family-level multimorbidity among urban (AOR: 1.642 (1.526–1.767)), southern region (AOR: 2.727 (2.467–3.015)) and the most affluent (AOR: 2.543 (2.284–2.832)) groups (Table 4). The various correlates observed for higher odds of conjugal multimorbidity were urban (AOR: 1.71 (1.58–1.84)), southern region (AOR: 2.64 (2.37–2.94)), families with four or more members (AOR: 1.21 (1.11–1.33)), and the most affluent class (AOR: 2.46 (2.19–2.76)). Multimorbidity at the level of siblings was associated with the southern region (AOR: 9.12 (3.29–25.29)), whereas intergenerational multimorbidity was associated with urban (AOR: 1.42 (1.17–1.72)), the southern part of India (AOR: 2.40 (1.82–3.17)) and the most affluent (AOR: 2.90 (2.18–3.87)) groups.

Table 5 suggests an increasing trend in healthcare expenditure (in both IPD and OPD cost) with an increase in the number of members with multimorbidity in a family. The median IPD cost for the families without multimorbid members was approximately INR 6900, which escalated to INR 20000 among families which had three or more multimorbid members. The OPD cost for the families with three or more multimorbid members raised to INR 1807 from INR 700 for families without multimorbid individuals.

## 4. Discussion

This study examined multimorbidity at the family level using data from 22,526 households. We observed that 44.6% of the households had family-level multimorbidity. The prevalence of multimorbidity among conjugal, sibling, and intergenerational pairs was almost similar. The major predictors of family-level multimorbidity were residing in an urban area, living in southern India, and being a part of the affluent class. We observed an increase in healthcare expenditure for both IPD and OPD, with an increase in the number of multimorbid members in a family.

We observed a high prevalence of family-level multimorbidity among older adults in India, which is in harmony with the findings of a previous study which reported coexistence of chronic pain, cardiovascular disease, and depression across family members after adjusting for known environmental risk factors [27]. Our findings suggest an urgent need for family-level interventions to combat multimorbidity. This accentuates the need for home-based practices such as dietary modifications and physical exercise, which may have an equal effect on a family’s health [28]. We explored the intermediaries of multimorbidity from a novel lens of looking at family as a unit rather than as individuals, which being a newer concept, limits our ability to compare our findings with the literature.

We observed an almost similar prevalence of multimorbidity among conjugal, sibling, and intergenerational pairs, which signifies that the environmental factors have an equal role in contributing to the long-term conditions rather than only genetic predispositions. Conjugal pairs which are not genetically related but had a high prevalence of multimorbidity which cannot be overlooked. Additionally, conjugal multimorbidity is important in countries such as India where predetermined gender roles determine the health outcomes of an individual [29]. Women often are not the decision makers in the household, which could be attributed to socioeconomic and cultural barriers [30]. Moreover, if both the partners have concordant conditions, it would be easier to coordinate routine check-ups and the management of medicines. However, in cases of discordant conditions, maintaining routine visits might become a challenge. Hence, future studies should explore disease clusters among conjugal pairs, which would help in planning strategies for setting care coordination goals.

Furthermore, sibling and intergenerational pairs with multimorbidity may have a genetic predisposition along with the environmental influence. However, within a family one person’s health also has an equal effect on other members. For example, if the bread winner of a family has multimorbidity, the other members will be concerned, which in turn could affect their mental health. In this context, Christensen proposed a conceptual framework named the “health-promoting family” in 2004, which laid stress on how the family can promote health and build their children’s capacity as health-promoting actors [31]. Further, it emphasised the role of environmental factors such as income, education, and resources, along with a family’s ecocultural pathway and practices including diet, exercise, and risk behaviours in shaping the wellbeing of the family. Additionally, a scoping review conducted to identify conceptual models of the health-promoting family recognised the child as an active member in promoting health of their family [13].

We found multimorbidity to be associated with urban residents, affluent groups, and Southern India, which is similar with the findings of previous studies conducted among individual-level participants in India [32,33]. Although these findings suggest these groups to have more multimorbidity, it could be due to various other social factors. Since our data was based on self-reported chronic conditions, urban residents may have better and improved access to healthcare facilities that could lead to better diagnosis of conditions in this group. A probable reason for higher multimorbidity among affluent groups could be that they received a better diagnosis due to their ability to pay as compared to the deprived groups. However, a systematic review conducted to investigate the associations between household- and area-level social determinants of health and multimorbidity found multimorbidity to be significantly associated with lower socioeconomic status or area-level deprivation [34]. In India, health is a state subject where the central government could give directions, but the ultimate powers to implement the programs lie with the states. Southern India has a well-developed public health system as compared to other parts of the country, which could be a reason for better diagnosis, and hence, higher reporting of multimorbidity in this region. Another reason could be the higher overall socioeconomic development and increased literacy rates in Southern India, which could have led to a higher use of healthcare services, and hence, better diagnosis of chronic conditions and multimorbidity.

We observed an increase in healthcare expenditure on both IPD and OPD visits with an increase in the number of members having multimorbidity. Furthermore, these challenges are exacerbated among the deprived groups where prioritisation amongst the family members would be required depending on the capacity of the family to pay. Our previous study among the urban poor in India showed a significantly higher healthcare expenditure on multimorbidity than those that did not had multimorbidity (no morbidity or one chronic condition) [35]. Here, it is worth noting that the ability to pay might further inhibit women and older adults from seeking care, as the priority would be given to the male and earning members of the family. Hence, strengthening primary care to provide egalitarian and responsive quality services is required. Public-funded primary care services are free in India, which if strengthened, can reduce a lot of out-of-pocket expenditure, thus preventing people from impoverishment.

### 4.1. Implications for Policy and Practice

This study points towards health promotion at the family level, which could be achieved through positive engagement in home-based practices. Educating women on health matters by using ongoing platforms such as Village Health and Nutrition Day (VHND) will be useful for empowering women to influence their family’s health choices. Additionally, women are often responsible for taking care of the family’s food and nutrition; thus, if empowered, they can directly influence their family’s dietary patterns [36]. Life course perspective is an important concept in family-level interventions as habits such as physical exercise and dietary intake developed at an early age will firmly stay throughout the life span of an individual [37]. This will also help in causing a reciprocal impact on other family members. Additionally, family communication can be a strong tool in the coping process [38]. Meaningful social connections along with family adjustment and adaptation should be promoted for holistic wellbeing. Although the findings suggest urban residents and affluent groups have a higher prevalence of multimorbidity, the focus should be centred on planning interventions for everyone as urban residents might have access to easy and improved healthcare facilities, causing our data to show a higher association of multimorbidity in this group. Similarly, affluent groups may easily seek care and diagnosis due to their ability to pay more, and hence, may report higher morbidity in our data. Nonetheless, the newly established Health and Wellness Centres (HWCs) envisage strengthening primary care by providing a basket of free preventive and curative services, which could be the new epicentre for providing a continuum of care to families.

### 4.2. Strengths and Limitations

This study introduced the concept of looking at the family as a unit rather than the individual in the context of multimorbidity. We used a large nation-wide dataset to generate evidence. Additionally, household-level potential determinants were used to explore the correlates of multimorbidity. However, all variables used were not collated at the family level; therefore, only limited independent variables could be used. Our study was exploratory in nature, and hence, we did not perform multilevel modelling but rather we extrapolated a few of the individual-level variables on the family level, which is another limitation. Additionally, we did not conduct a weighted analysis as only the individual-level weight was available from the LASI. The chronic conditions included were self-reported, which could have undermined the true prevalence. This study is based on cross-sectional data, thus limiting the establishment of causality. We did not perform modelling of healthcare expenditure variables; however, future studies should investigate this to understand how family-level multimorbidity impacts it. However, to compensate for this we used the median instead of the mean to present the results.

## 5. Conclusions

This study reports on the novel concept of looking at multimorbidity by considering the family or household as a unit. Multimorbidity was found to be increasingly prevalent in the family; thus, necessitating the need for family-centred intervention. Additionally, we not only observed that genetically related individuals had a higher prevalence of multimorbidity, but prevalence was also high amongst conjugal pairs, which signifies that the shared risk factors beyond genetics should equally be focused on to mitigate multimorbidity. Future studies should explore intermediaries of family-level multimorbidity through a syndemic lens.

## Figures and Tables

**Figure 1 ijerph-19-09850-f001:**
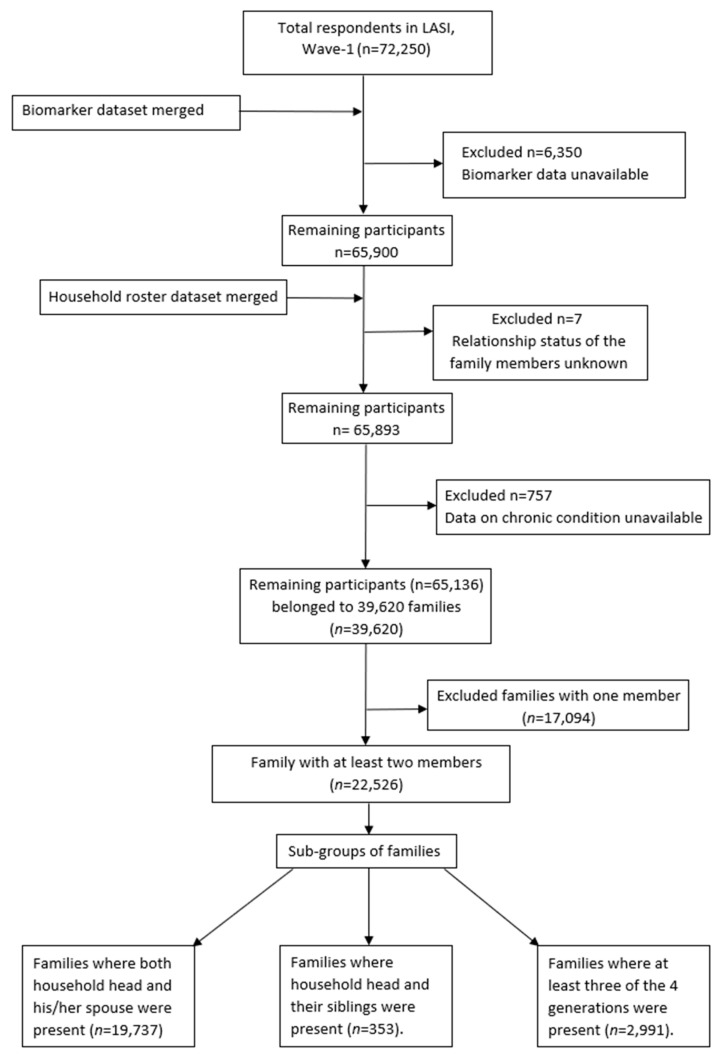
Selection of study participants. LASI compiled data into three separate datasets, namely, “individual”, “biomarker”, and “household roster” datasets, which were merged to create various variables used in the present study.

**Table 1 ijerph-19-09850-t001:** Sociodemographic characteristics of the study population (*n* = 22,526 families).

Household Characteristics		*n* (%)
Household average age (*n* = 22,526)		Mean: 56.81 (±9.15) years Range: 33–97.5 years
Residence (*n* = 22,358)		
	Rural	14,781 (66.11)
	Urban	7577 (33.89)
Household average education		Mean: 7.64 (±3.42) years Range: 1–20.5 years
Caste (*n* = 22,358)		
	Scheduled Caste (SC)	3743 (16.74)
	Scheduled Tribe (ST)	3924 (17.55)
	Other Backward Class (OBC)	8557 (38.27)
	None of them	6134 (27.44)
Region of India (*n* = 22,358)		
	North	3249 (14.53
	Central	3822 (17.09)
	East	4227 (18.91)
	Northeast	3038 (13.59)
	West	2771 (12.39)
	South	5251 (23.49)
Household Size (*n* = 22,358)		
	No. of members	5.12 (±2.52)
Smoking inside household (*n* = 22,317)		
	Yes	6142 (27.52)
	No	16,175 (72.48)
Cooking fuel (*n* = 22,358)		
	Coal/wood/dung cake	10,280 (45.98)
	LPG^†^/biogas	11,776 (52.67)
	Other	302 (1.35)
Source of drinking water (*n* = 22,317)		
	Pipe/tap water	10,895 (48.82)
	Well	9499 (42.56)
	Other sources	1923 (8.62)
Household-level wealth index (*n* = 22,358)		
	Poorest	4258 (19.04)
	Poorer	4512 (20.18)
	Middle	4571 (20.44)
	Richer	4615 (20.64)
	Richest	4402 (19.69)

**Table 2 ijerph-19-09850-t002:** Proportion of multimorbidity at the family level (*n* = 22,526 families).

No. of Family Members	Number of Family Members with Multimorbidity				
	None	One	Two	Three	>Three
2 members	4949 (24.68)	6718 (33.50)	8389 (41.83)	-	-
3 members	280 (13.74)	478 (23.45)	640 (31.40)	640 (31.40)	-
4 members	20 (5.51)	56 (15.43)	87 (23.97)	94 (25.90)	106 (29.20)
>4 members	5 (7.25)	4 (5.80)	7 (10.14)	14 (20.29)	39 (56.52)

**Table 3 ijerph-19-09850-t003:** Prevalence of multimorbidity amongst various groups of household members.

Conjugal multimorbidity (household head and their spouse) (*n* ^#^ = 19,737 families with both spouses)	
No. of spouse(s) having multimorbidity	Multimorbidity status *n* (%)
None of the partners	4969 (25.18)
Either of the partners	6582 (33.35)
Both of the partners	8186 (41.48)
2. Multimorbidity amongst siblings of household head (*n* ^#^ = 353 families with ≥2 siblings)	
No. of sibling(s) having multimorbidity	Multimorbidity status *n* (%)
None	90 (25.49)
One	116 (32.86)
≥2	147 (41.64)
3. Intergenerational multimorbidity (*n* ^#^ = 2991 families with ≥2 generations)	
No. of member(s) having multimorbidity	Multimorbidity status *n* (%)
None	531 (17.75)
One	1082 (36.18)
≥2	1378 (46.07)

# *n* is the number of group-specific eligible families out of the total number of 22,526 families.

**Table 4 ijerph-19-09850-t004:** Association between family-level, conjugal, sibling, and intergenerational multimorbidity with various sociodemographic correlates.

Sociodemographic Characteristics	Categories	AOR (95% Confidence Interval)			
		Family-Level Multimorbidity	Conjugal Multimorbidity	Sibling Multimorbidity	Intergenerational Multimorbidity
Household average age		1.05 (1.047–1.055)	1.04 (1.04–1.05)	1.03 (0.99–1.06)	1.01 (1.00–1.03)
Residence	Rural	Reference			
	Urban	1.642 (1.526–1.767)	1.71 (1.58–1.84)	1.52 (0.78–2.96)	1.42 (1.17–1.72)
Household average education		1.046 (1.038–1.054)	1.03 (1.02–1.04)	1.09 (1.01–1.17)	1.04 (1.02–1.06)
Caste	ST *	Reference			
	SC ^	1.442 (1.288–1.615)	1.40 (1.24–1.59)	0.85 (0.30–2.35)	1.41 (1.03–1.93)
	OBC ^#^	1.437 (1.302–1.586)	1.37 (1.23–1.52)	1.51 (0.67–3.41)	1.53 (1.16–2.02)
	Other	1.796 (1.612–2.000)	1.72 (1.53–1.93)	1.89 (0.77–4.62)	1.99 (1.48–2.69)
Regions of India	Central	Reference			
	North	2.171 (1.943–2.426)	2.18 (1.94–2.46)	2.32 (0.73–7.32)	1.92 (1.42–2.60)
	East	1.781 (1.609–1.971)	1.70 (1.52–1.89)	2.71 (0.96–7.63)	1.63 (1.24–2.13)
	Northeast	1.142 (1.005–1.296)	1.12 (0.98–1.29	2.79 (0.78–9.92)	1.12 (0.78–1.62)
	West	1.858 (1.655–2.086)	1.86 (1.64–2.10)	1.83 (0.57–5.85)	1.41 (1.03–1.93)
	South	2.727 (2.467–3.015)	2.64 (2.37–2.94)	9.12 (3.29–25.29)	2.40 (1.82–3.17)
No. of household members	Two	0.717 (0.653–0.787)	1.08 (0.96–1.21)	0.42 (0.14–1.28)	0.65 (0.40–1.05)
	Three	0.730 (0.665–0.801)	1.04 (0.93–1.16)	0.91 (0.27–3.07)	0.73 (0.50–1.06)
	Four	0.742 (0.681–0.807)	1.21 (1.11–1.33)	0.65 (0.28–1.49)	1.04 (0.81–1.35)
	≥Five	Reference			
Smoking inside household	Yes	Reference			
	No	1.013 (0.945–1.084)	1.03 (0.96–1.11)	0.57 (0.30–1.066)	1.05 (0.87–1.27)
Cooking fuel	Coal/wood/dung cake	Reference			
	LPG ^†^/biogas	1.468 (1.366–1.578)	1.43 (1.33–1.55)	2.02 (1.06–3.82)	1.25 (1.03–1.51)
	Other	1.243 (0.942–1.640)	1.14 (0.85–1.55)	3.35 (0.25–44.91)	1.46 (0.63–3.40)
Source of drinking water	Pipe/tap water	Reference			
	Well	1.030 (0.961–1.105)	1.08 (1.01–1.17)	1.44 (0.80–2.59)	0.85 (0.70–1.02)
	Other sources	1.093 (0.979–1.221)	1.15 (1.02–1.29)	1.85 (0.62–5.46)	1.08 (0.79–1.47)
Household-level wealth index	Poorest	Reference			
	Poorer	1.331 (1.207–1.467)	1.32 (1.19–1.47)	2.15 (0.90–5.15)	1.38 (1.08–1.76)
	Middle	1.457 (1.321–1.607)	1.45 (1.30–1.61)	1.93 (0.79–4.67)	1.42 (1.11–1.83)
	Richer	2.018 (1.826–2.231)	2.04 (1.83–2.27)	1.80 (0.72–4.47)	1.76 (1.36–2.28)
	Richest	2.543 (2.284–2.832)	2.46 (2.19–2.76)	1.36 (0.49–3.81)	2.90 (2.18–3.87)

* ST: Scheduled Tribes; ^ SC: Scheduled Caste; # OBC: Other Backward Class; ^†^ LPG: liquefied petroleum gas.

**Table 5 ijerph-19-09850-t005:** Healthcare expenditure (in Indian national rupee) towards inpatient and outpatients services.

Household Out-of-Pocket Healthcare-Expenditure	Number of Household Members with Multimorbidity			
	None	One	Two	Three or More
	Median (IQR)	Median (IQR)	Median (IQR)	Median (IQR)
IPD cost (INR)	6900 (2000–22,000)	9000 (3000–28,000)	11,750 (3500–34,660)	20,000 (6100–50,000)
OPD cost (INR)	700 (300–1560)	900 (400–2000)	1163 (500–2650)	1807 (880–4000)

## Data Availability

The dataset analysed during the current study is available in the LASI data repository held at ICT, IIPS (https://g2aging.org/?section=overviews&study=lasi, accessed on 15 December 2021).

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
