# Peer review of "Family-Level Multimorbidity among Older Adults in India: Looking through a Syndemic Lens"

_ijerph, 2022, doi:10.3390/ijerph19169850_

Round 1

Reviewer 1 Report

Dear editor,

The paper discusses an issue that is both interesting and has important policy ramifications. Few publications examined the relationship between multimorbidity and families. The authors could make reference to van Hecke (2017) in this regard. However the reading doesn't always flow easily due to the language and a number of omissions that appear to have been made in a rush to finish the article.

Three are my main concerns.  Even though the authors state in the introduction that they would examine two novel ideas, "family level multimorbidity" and "family level complicated multimorbidity," in actuality only the first is covered in great detail. For the second, which is just addressed in section 4, there is no outcome displayed. Then, a paragraph outlining the rationale for selecting particular sociodemographic and economic factors is entirely missing. The authors can also cite Ingram et al. 2021, Singer et al. 2019, Lucyk et al. 2017, and Agborsangaya et al. 2012 in this regard. In addition there is no performance test conducted in conjunction with the estimations.

A more thorough explanation of the data collected by the LASI survey and the authors' use of it would enhance Section 2. In particular, a description of the procedures followed and the information used, as shown in Fig. 1, may be sufficient. Table 2's "No of spouses” report that they can be: none, either, or both. This appears to indicate that a head of home can have a maximum of two wives. This kind of information ought to be included in a more thorough description of the sample being studied.

To improve the style, eliminate extra words, prevent leaving out important details, and locate more appropriate terms, the paper should be carefully read or proofread. For instance:

-row 19 the authors refer to two new definitions but only one is listed

-row 51 replace “conjugal” with couples and “sibling” with siblings

- row 44 “to combat” can be deleted

-row 37 consider revising the following sentence“without defining index disease”

- row 119 can be put in a footnote. 

For the reasons previously expressed I think that the paper can be published after minor revisions.

Author Response

Reviewer 1

Dear editor,

The paper discusses an issue that is both interesting and has important policy ramifications. Few publications examined the relationship between multimorbidity and families. The authors could make reference to van Hecke (2017) in this regard.

Author’s Response: Thank you so much for your kind comments. As suggested we have cited van Hecke (2017) in the manuscript.

However the reading doesn't always flow easily due to the language and a number of omissions that appear to have been made in a rush to finish the article.

Author’s Response: We have thoroughly edited the manuscript for grammatical errors and hope it now seems to be fine.

Three are my main concerns.  Even though the authors state in the introduction that they would examine two novel ideas, "family level multimorbidity" and "family level complicated multimorbidity," in actuality only the first is covered in great detail. For the second, which is just addressed in section 4, there is no outcome displayed.

Author’s Response: We agree with the reviewer and following their recommendation we have now completely removed the concept of family level complex multimorbidity from the manuscript.

Then, a paragraph outlining the rationale for selecting particular sociodemographic and economic factors is entirely missing. The authors can also cite Ingram et al. 2021, Singer et al. 2019, Lucyk et al. 2017, and Agborsangaya et al. 2012 in this regard. In addition there is no performance test conducted in conjunction with the estimations.

Author’s Response: Thank you for your suggestion. We have added the paragraph and all suggested literature have now been cited.

A more thorough explanation of the data collected by the LASI survey and the authors' use of it would enhance Section 2. In particular, a description of the procedures followed and the information used, as shown in Fig. 1, may be sufficient.

Author’s Response: We have done the changes as suggested.

Table 2's "No of spouses” report that they can be: none, either, or both. This appears to indicate that a head of home can have a maximum of two wives. This kind of information ought to be included in a more thorough description of the sample being studied.

Author’s Response: No of spouse means between the couple, neither husband nor wife; or either husband or wife; or both husband and wife have multimorbidity. We have now edited it and also mentioned few lines to make this statement more clear.

To improve the style, eliminate extra words, prevent leaving out important details, and locate more appropriate terms, the paper should be carefully read or proofread. For instance:

-row 19 the authors refer to two new definitions but only one is listed

Author’s Response: Done as suggested.

-row 51 replace “conjugal” with couples and “sibling” with siblings

Author’s Response: Sibling has been changed to siblings as suggested. However, we chose to keep conjugal word in line with most of the scientific literature using this word over couples. 

- row 44 “to combat” can be deleted

Author’s Response: We have now used the term ‘mitigate’ instead of combat and have further modified the sentence.

-row 37 consider revising the following sentence “without defining index disease”

Author’s Response: Done as suggested.

- row 119 can be put in a footnote.                       

Author’s Response: Done as suggested.

For the reasons previously expressed I think that the paper can be published after minor revisions.

Author’s Response: We thank you again for your valuable suggestions and time.

Reviewer 2 Report

Abstract: 

1. "We propose two new terms": I can only see one: "family level multimorbidity". 

2. "Multivariate logistic regression assessed correlates": The outcome of interest is unclear. Is it healthcare expenditure or family multimorbidity? maybe both?

3. "41.8% (both partners) had conjugal multimorbidity": I would recommend moving "(both partners)" after "conjugal". 

4. I think the sentence: "We report the novel concept of looking at multimorbidity by considering family or household as a unit." could be deleted as that was said already earlier in the abstract. 

5. The implications of this work are unclear: "Our findings necessitate family centred intervention rather than individualized approach". How was this conclusion reached based on the results of the regression model? I am struggling to follow the last sentence. What are some examples of intermediaries of family level multimorbidity through syndemic lens? Simpler phrasing may help here. 

Introduction:

1. The sentence "Our recent study conducted in healthcare facilities of Odisha"... : Maybe add "in Eastern India, Odisha" for those not familiar with India's geography and the sample size used to generate the 55% multimorbidity prevalence estimate. 

2. "Impeding the provision of quality of care": I think an adjustive such as  "appropriate" or "good" should be added to quality of care.

3. Line 43-44 seems incomplete (to combat what?). Also, the meaning of "policy level interventions" is unclear.

4. Lines 45-46: This sentence could be enhanced by some rephrasing. Maybe starting with "In India, around 77%...". 

5. Line 56: Do you mean "successful" rather than "sustainable"?

6. Line 57: I would remove the "Additionally". 

7. Lines 66 and 67: Are there any references to support this statement? Any evidence from single diseases and family-level interventions?

8. Lines 68-70:  When stating the study goal, "family level complex multimorbidity" is not mentioned. How this concept is incorporated into the study is unclear.

Material and Methods

1. Lines 74-81: I think this paragraph could be rephrased so it's clearer that the authors were not involved in data collection and used an already collected, available, dataset. Line 80: "Following way" is unclear. Also, how did the sample decrease from 72,250 to 42,949?

2. Line 91: Is age>=45 an additional exclusion criteria? Or are all respondents of LASI already 45+?

3. Figure 1:

-What are the "Biomarker" and "Household roster" datasets? Are they different parts/modules of LASI? A note at the bottom of the figure would help.

-The sub-groups of families (3 boxes at the very bottom) do not add up to 22,526. Is that because some families were in more than one category? If so, that should be clarified. Maybe also as a note if not in the main text.

4. Lines 107-109: Please define OPD and IPD. Did all costs correspond to the same year (if so which)? Was this variable self-reported? If so, does it only capture out of pocket expenditure for the families or does it also capture costs of healthcare provision? Also, isn't healthcare expenditure an outcome measure? Based on the study aims, it seemed so.

5. Lines 111-117: Please rephrase as the sentence is incomplete and hard to follow. Maybe start with sentence 117-118 and then move to the list of conditions. 

6. Lines 119-120: Based on the study aims, isn't BMI conceptualized as an independent variable rather than an outcome of interest? Also, I didn't see it included in the regression model presented later on.

7. Line 122: I would recommend moving the description of the statistical software to the end of the statistical analysis section. 

8. Lines 122-123:  In my opinion, "descriptive statistics" is a more commonly used term than "descriptive estimation" as the latter seems to imply that a  regression model was used to adjust for confounders.

9. Lines 124-125: I would recommend stating "for categorical variables" rather than "attribute" as I think the authors are presenting the types of descriptive statistics used for each type of variable (continuous vs. categorical) in the dataset. 

9. Line 126: Please define acronyms the first time they are used (OPD and IPD are first mentioned in the outcomes section).

10.  Line 128: I think it would help to state the variables that were adjusted for in the regression models. Those would have been described in the previous section "independent variables" so only listing them should suffice here to clarify the structure of the regression model. Also, a rationale for which variables were included in the model is missing (how were independent variables chosen?). 

11. How healthcare expenditure was analysed is missing from the statistical analysis section. 

12. How was the goodness of fit of the models evaluated?

Results

1. Line 139: What type of descriptive statistics is "+-2.52"? Please clarify and include in brackets.

2. Table 1: I think the clarity of this table could be improved by re-labeling the column "Multimorbidity" into "No. of Family Members with Multimorbidity". I also wonder if most of the information in Table S2 could be integrated into Table 1.

3. Lines 160: I think Odds Ratio (OR) rather than "AOR" is the convention in most journals, as the previous sentence clearly states that these are adjusted ORs from a regression model. 

4. Table 3. Please define any acronyms used in the notes section of the table. What do ST, SC, OBC, and LPG mean?

5. I think in Table 3 it is the first time where the variables controlled for in the regression model are listed. As indicated in a previous comment, please specify the variables the model was adjusted for in the methods section. Also, several variables presented in the table were not previously defined in the methods section (independent variables). 

6. Table 3. I would recommend changing "AOR" for "Odds Ratio".

7. Table 3. I would have liked to see descriptive statistics on the variables listed in this table. What was the percentage of missing values for each? The manuscript may benefit for a slight change in tables. Table 1 could present the sample characteristics. Table 2 could be a more concise version of current Tables 1 and 2. And then Table 3 would follow as is. 

8. Line 172. I would recommend a change in the wording into "suggests" or "indicates" rather than "evidently establishes". 

9. Table 4. Please add the year that health expenditures correspond too. As previously mentioned, the analyses on healthcare expenditure were not detailed in the methods section. Also, I think the statistics presented in Table 4 are unadjusted results, correct? Did the authors consider calibrating a regression model that adjusts for differences in sociodemographic variables that may be confounding the health expenditure results presented in Table 4?

Discussion

Line 188-189. The prevalence of family level complex multimorbidity was close to 18%, which I wouldn't consider as "low".

Line 189-190: I am not sure this statement can be drawn based on the analyses provided in this study. This study indicates that family level multimorbidity is relatively prevalent (45%), but the specific disease profile (and age or gender) of each family member may be different and, thus, may still benefit from an individualised approach. Instead, the authors may consider indicating that interventions aimed at reducing multimorbidity may also consider if multimorbidity is common at the family level. Were the authors able to identify the percentage of individuals with multimorbidity who also had family-level multimorbidity versus those with multimorbidity but not at the family level? 

Lines 192-195: I don't think these lines follow from the study findings. I think a 18% of households with complex multimorbidity is quite high actually and deserves further attention. 

Lines 195-197: These lines do not seem to belong here and the authors may consider the possibility of moving to a specific section on strengths and limitations. 

Lines 227-229: I don't quite follow these sentences. I think "Universal health coverage" is more commonly used at the system level, to refer to the healthcare system of a country. Also, the sentence on primary care does not follow from the previous statement. Is primary care free for all citizens in India? A few sentences describing the healthcare system in India (and whether there are any out of pocket costs for users) would be helpful. 

Implications for policy and practice

1. Line 232: What does "positive engagement" and "individual responsiveness" mean?

2. Lines 232-234: Doesn't this statement contradict the previous sentence: "Women often are not the decision makers in the household which could be attributed to the socio-economic and cultural barriers" ?

3. Line 237: Please provide another example rather than "etc."

4. Line 242: Isn't it a bit counterintuitive to suggest that interventions should focus on the most affluent? Wouldn't that further exacerbate health inequalities? The authors may want to hypothesize a bit more on why higher multimorbidity may be present among the most affluent. In general, more details on the interpretation of the findings also for urban and southern India would be helpful. 

5. Strengths and limitations: What about any limitations in measuring health expenditure? 

6. By reading the limitation on data NOT collected at the family level, I wonder why the authors did not use a multilevel model rather than aggregating all variables a the family level. 

Conclusion

1. Please remove the comma after "This study". 

2. I don't quite agree with the conclusion of thinking about family and individual-level approaches as substitutes. I think this study points at the possibility of also considering the family, but provides no evidence on the inferiority of individual-level interventions. 

Author Response

Reviewer 2

Abstract: 

  1. "We propose two new terms": I can only see one: "family level multimorbidity". 

Author’s Response: We have kept it as one term now. Thank you for pointing this out.

  1. "Multivariate logistic regression assessed correlates": The outcome of interest is unclear. Is it healthcare expenditure or family multimorbidity? maybe both?

Author’s Response: Our primary aim is to estimate the prevalence and correlates of family level multimorbidity followed by secondary aim to assess healthcare expenditure.

  1. "41.8% (both partners) had conjugal multimorbidity": I would recommend moving "(both partners)" after "conjugal". 

Author’s Response: Done as suggested.

  1. I think the sentence: "We report the novel concept of looking at multimorbidity by considering family or household as a unit." could be deleted as that was said already earlier in the abstract. 

Author’s Response: Done as suggested.

  1. The implications of this work are unclear: "Our findings necessitate family centred intervention rather than individualized approach". How was this conclusion reached based on the results of the regression model? I am struggling to follow the last sentence. What are some examples of intermediaries of family level multimorbidity through syndemic lens? Simpler phrasing may help here. 

Author’s Response: Thank you for your suggestion. We have now revised it to “Our findings depict family centred interventions may be considered to mitigate multimorbidity. Future studies should explore family level multimorbidity to help inform programs and policies in strategizing preventive as well as curative services with family as a unit.”

Introduction:

  1. The sentence "Our recent study conducted in healthcare facilities of Odisha"... : Maybe add "in Eastern India, Odisha" for those not familiar with India's geography and the sample size used to generate the 55% multimorbidity prevalence estimate. 

Author’s Response: We feel nationally representative data can better highlight the magnitude of multimorbidity in India and hence we have changed this sentence.

  1. "Impeding the provision of quality of care": I think an adjustive such as  "appropriate" or "good" should be added to quality of care.

Author’s Response: Done as suggested.

  1. Line 43-44 seems incomplete (to combat what?). Also, the meaning of "policy level interventions" is unclear.

Author’s Response: Line rephrased to make it clearer.

  1. Lines 45-46: This sentence could be enhanced by some rephrasing. Maybe starting with "In India, around 77%...". 

Author’s Response: Done as suggested.

  1. Line 56: Do you mean "successful" rather than "sustainable"?

Author’s Response: We have now changed it to successful. Thank you for your suggestion.

  1. Line 57: I would remove the "Additionally". 

Author’s Response: Done as suggested.

  1. Lines 66 and 67: Are there any references to support this statement? Any evidence from single diseases and family-level interventions?

Author’s Response: We now state that it ‘may’ be useful as there are no references to support this.

  1. Lines 68-70:  When stating the study goal, "family level complex multimorbidity" is not mentioned. How this concept is incorporated into the study is unclear.

Author’s Response: We have removed the concept of family level complex multimorbidity now in line with the suggestions of reviewers.

Material and Methods

  1. Lines 74-81: I think this paragraph could be rephrased so it's clearer that the authors were not involved in data collection and used an already collected, available, dataset. Line 80: "Following way" is unclear. Also, how did the sample decrease from 72,250 to 42,949?

Author’s Response: We have now rephrased this sentence. Additionally, we want to specify that 72250 individuals were sampled from 42949 households. In the present study we have considered unit of analysis as household.

  1. Line 91: Is age>=45 an additional exclusion criteria? Or are all respondents of LASI already 45+?

Author’s Response: As already mentioned in the manuscript, LASI included all adults aged 45 years and above in a household along with their spouse who may be of any age. Thus, participants who are non-representative of sample (less than 45 years) are there who were excluded based on the inclusion criteria of ≥45 years.

  1. Figure 1:

-What are the "Biomarker" and "Household roster" datasets? Are they different parts/modules of LASI? A note at the bottom of the figure would help.

Author’s Response: The LASI data is available as different datasets which could be merged with each other. The three available datasets i.e. individual, biomarkers and household were used for this study after merging them.

-The sub-groups of families (3 boxes at the very bottom) do not add up to 22,526. Is that because some families were in more than one category? If so, that should be clarified. Maybe also as a note if not in the main text.

Author’s Response: Yes, some families are in more than one category. We have now mentioned this as suggested by the reviewer.

  1. Lines 107-109: Please define OPD and IPD. Did all costs correspond to the same year (if so which)? Was this variable self-reported? If so, does it only capture out of pocket expenditure for the families or does it also capture costs of healthcare provision? Also, isn't healthcare expenditure an outcome measure? Based on the study aims, it seemed so.

Author’s Response: All abbreviations have been defined at first use now. Yes, all costs correspond to same year as LASI is a cross-sectional study (now mentioned in the manuscript). It captures only out of pocket expenditure. Our primary aim is to estimate the prevalence and correlates of family level multimorbidity followed by secondary aim to assess healthcare expenditure.

  1. Lines 111-117: Please rephrase as the sentence is incomplete and hard to follow. Maybe start with sentence 117-118 and then move to the list of conditions. 

Author’s Response: This sentence has been rephrased as suggested.

  1. Lines 119-120: Based on the study aims, isn't BMI conceptualized as an independent variable rather than an outcome of interest? Also, I didn't see it included in the regression model presented later on.

Author’s Response: BMI is considered as an individual chronic condition based on the new literature and hence has been used to assess multimorbidity. It is included in 17 chronic conditions. Please refer to this:

Burki T. European Commission classifies obesity as a chronic disease. The Lancet Diabetes & Endocrinology. 2021 Jul 1;9(7):418.

  1. Line 122: I would recommend moving the description of the statistical software to the end of the statistical analysis section. 

Author’s Response: Done as suggested.

  1. Lines 122-123:  In my opinion, "descriptive statistics" is a more commonly used term than "descriptive estimation" as the latter seems to imply that a regression model was used to adjust for confounders.

Author’s Response: Done as suggested.

  1. Lines 124-125: I would recommend stating "for categorical variables" rather than "attribute" as I think the authors are presenting the types of descriptive statistics used for each type of variable (continuous vs. categorical) in the dataset. 

Author’s Response: Done as suggested.

  1. Line 126: Please define acronyms the first time they are used (OPD and IPD are first mentioned in the outcomes section).

Author’s Response: Done as suggested.

  1. Line 128: I think it would help to state the variables that were adjusted for in the regression models. Those would have been described in the previous section "independent variables" so only listing them should suffice here to clarify the structure of the regression model. Also, a rationale for which variables were included in the model is missing (how were independent variables chosen?). 

Author’s Response: Done as suggested.

  1. How healthcare expenditure was analysed is missing from the statistical analysis section. 

Author’s Response: It is already mentioned in the manuscript. However, we have now added few more lines on it.

  1. How was the goodness of fit of the models evaluated?

Author’s Response: It is now mentioned in the manuscript. R2 was used for it.

Results

  1. Line 139: What type of descriptive statistics is "+-2.52"? Please clarify and include in brackets.

Author’s Response: Done (mean±sd)

  1. Table 1: I think the clarity of this table could be improved by re-labeling the column "Multimorbidity" into "No. of Family Members with Multimorbidity". I also wonder if most of the information in Table S2 could be integrated into Table 1.

Author’s Response: We have relabelled as suggested by the reviewer. However, we chose to keep the tables as it is for their better clarity. We hope the reviewer would understand this.

  1. Lines 160: I think Odds Ratio (OR) rather than "AOR" is the convention in most journals, as the previous sentence clearly states that these are adjusted ORs from a regression model. 

Author’s Response: Although, we agree with the reviewer that stating OR would equally be good instead of AOR. However, we see AOR more commonly used in literature and therefore, we chose to keep the latter.

  1. Table 3. Please define any acronyms used in the notes section of the table. What do ST, SC, OBC, and LPG mean?

Author’s Response: Done as suggested.

  1. I think in Table 3 it is the first time where the variables controlled for in the regression model are listed. As indicated in a previous comment, please specify the variables the model was adjusted for in the methods section. Also, several variables presented in the table were not previously defined in the methods section (independent variables). 

Author’s Response: It has now been specified in analysis section also.

  1. Table 3. I would recommend changing "AOR" for "Odds Ratio".

Author’s Response: We thank the reviewer for suggestion. However, for the reasons stated previously, we chose to keep it as AOR.

  1. Table 3. I would have liked to see descriptive statistics on the variables listed in this table. What was the percentage of missing values for each? The manuscript may benefit for a slight change in tables. Table 1 could present the sample characteristics. Table 2 could be a more concise version of current Tables 1 and 2. And then Table 3 would follow as is. 

Author’s Response: We have now added Table 1 as suggested. However, we chose to keep other tables as it is. Merging them may lose their clarity. We thank reviewer for their suggestion.

  1. Line 172. I would recommend a change in the wording into "suggests" or "indicates" rather than "evidently establishes". 

Author’s Response: Done as suggested.

  1. Table 4. Please add the year that health expenditures correspond too. As previously mentioned, the analyses on healthcare expenditure were not detailed in the methods section. Also, I think the statistics presented in Table 4 are unadjusted results, correct? Did the authors consider calibrating a regression model that adjusts for differences in socio-demographic variables that may be confounding the health expenditure results presented in Table 4?

Author’s Response: We have now added year along with detailed description of the expenditure variables. Yes, these are unadjusted statistics. This study is an exploratory study where we introduced a new concept. Additionally, data on expenditure is skewed due to which we could present it as median only.

Discussion

Line 188-189. The prevalence of family level complex multimorbidity was close to 18%, which I wouldn't consider as "low".

Author’s Response: We have now removed this concept.

Line 189-190: I am not sure this statement can be drawn based on the analyses provided in this study. This study indicates that family level multimorbidity is relatively prevalent (45%), but the specific disease profile (and age or gender) of each family member may be different and, thus, may still benefit from an individualised approach. Instead, the authors may consider indicating that interventions aimed at reducing multimorbidity may also consider if multimorbidity is common at the family level. Were the authors able to identify the percentage of individuals with multimorbidity who also had family-level multimorbidity versus those with- multimorbidity but not at the family level? 

Author’s Response: We have now changed the statement as suggested by the reviewer. Yes, we have identified the percentage of individuals with multimorbidity who also had family-level multimorbidity versus those with- multimorbidity but not at the family level as presented in Table 2 and supplementary table S1.

Lines 192-195: I don't think these lines follow from the study findings. I think a 18% of households with complex multimorbidity is quite high actually and deserves further attention. 

Author’s Response: The concept of complex multimorbidity has now been removed.  

Lines 195-197: These lines do not seem to belong here and the authors may consider the possibility of moving to a specific section on strengths and limitations. 

Author’s Response: Although, it can be moved to limitations section. However, this fits in the context here and hence, we chose to highlight it here.  

Lines 227-229: I don't quite follow these sentences. I think "Universal health coverage" is more commonly used at the system level, to refer to the healthcare system of a country. Also, the sentence on primary care does not follow from the previous statement. Is primary care free for all citizens in India? A few sentences describing the healthcare system in India (and whether there are any out of pocket costs for users) would be helpful. 

Author’s Response:  We have now modified with in line with the suggestions of the reviewers.  

Implications for policy and practice

  1. Line 232: What does "positive engagement" and "individual responsiveness" mean?

Author’s Response: We have now modified this statement for better clarity and understand-ability.

  1. Lines 232-234: Doesn't this statement contradict the previous sentence: "Women often are not the decision makers in the household which could be attributed to the socio-economic and cultural barriers" ?

Author’s Response: Thank you for the suggestion. We have now modified the statement to make it more clear. However, both statements stand true in Indian context.

  1. Line 237: Please provide another example rather than "etc."

Author’s Response: Revised as suggested with another example.

  1. Line 242: Isn't it a bit counterintuitive to suggest that interventions should focus on the most affluent? Wouldn't that further exacerbate health inequalities? The authors may want to hypothesize a bit more on why higher multimorbidity may be present among the most affluent. In general, more details on the interpretation of the findings also for urban and southern India would be helpful. 

Author’s Response: Thank you for the suggestion. We have now highlighted and discussed why most affluent may have more multimorbidity. Additionally, in the interpretation of findings we have added urban, South India also. 

  1. Strengths and limitations: What about any limitations in measuring health expenditure? 

Author’s Response: Thank you for pointing this out. We have now added this.

  1. By reading the limitation on data NOT collected at the family level, I wonder why the authors did not use a multilevel model rather than aggregating all variables a the family level. 

Author’s Response: This is an exploratory study for introducing a new concept. Hence, we tried to give an overview of family as a unit of analysis. However, as suggested by the reviewer we have now mentioned it as a limitation.

Conclusion

  1. Please remove the comma after "This study". 

Author’s Response: Done as suggested.

  1. I don't quite agree with the conclusion of thinking about family and individual-level approaches as substitutes. I think this study points at the possibility of also considering the family, but provides no evidence on the inferiority of individual-level interventions. 

Author’s Response: Thank you for pointing this out. We have now modified the conclusion. The present conclusion is in line with the suggestions of reviewer.

We thank the reviewer for their valuable suggestions and time.

Reviewer 3 Report

In my opinion the research work is valuable and novel and deals with a topic of great interest. I believe that the work is well founded and structured and the methodological approach is adequate. However, I believe that the research work could be improved if some aspects were included, especially in the literature review. In recent years the concept of catastrophic health expenditure is commonly used and I think that mention should have been made of this concept and of some studies that have been carried out in recent years on this topic in some countries such as Brazil, USA or China (e.g. https://doi.org/10.11606/s1518-8787.2020054002285 or doi:10.1001/jamanetworkopen.2022.14923 ). I also think it would be desirable to refer to a recent research work on multimorbidity in India and poverty https://doi.org/10.3389/fpubh.2022.881967. 

Author Response

Reviewer 3

In my opinion the research work is valuable and novel and deals with a topic of great interest. I believe that the work is well founded and structured and the methodological approach is adequate. However, I believe that the research work could be improved if some aspects were included, especially in the literature review. In recent years the concept of catastrophic health expenditure is commonly used and I think that mention should have been made of this concept and of some studies that have been carried out in recent years on this topic in some countries such as Brazil, USA or China (e.g. https://doi.org/10.11606/s1518-8787.2020054002285 or doi:10.1001/jamanetworkopen.2022.14923 ). I also think it would be desirable to refer to a recent research work on multimorbidity in India and poverty https://doi.org/10.3389/fpubh.2022.881967. 

Author’s Response: Thank you so much for your kind consideration. We have mentioned catastrophic health expenditure and cited all the literature as suggested by the reviewer. We are happy that the reviewer has taken a keen interest in our previous work on multimorbidity in India and poverty.

We again thank the reviewer for their valuable suggestions and time. 

Round 2

Reviewer 2 Report

1. Line 64. I think you meant "continuity" of care.

2. I still find missing a couple of sentences describing the healthcare system in India. Do people have to only pay copayments or are there other types of out-of-pocket expenditure?

3. I think listing health expenditure under the independent variables section is strange as it seems to imply that you're predicting multimorbidity based on it.  I think it belongs to the outcomes of interest section. 

4. Line 165: a space is missing between "variables" and "adjusted".

5. The R-squared of the final model seems to be missing. 

6. Line 284 is incomplete.

7. Line 289"higher healthcare expenditure on multimorbidity" is a bit hard to understand. Higher than what?

8. "A study conducted in USA revealed a significant decrease in catastrophic health expenditure on multimorbidity among households between 2008 and 2018" It seems like the reason for the decline is unclear. What that due to the ACA? Also, do the authors think that the Indian and USA healthcare systems are comparable?

9. The skewness of healthcare expenditure data is a known property and I wouldn't state it as a particular limitation of your study. I think, instead, you may want to highlight that no modelling of healthcare expenditure variable was done, to understand, for example, how family multimorbidity impacts it. And how further research may want to look at that.

Author Response

Reviewer 2

  1. Line 64. I think you meant "continuity" of care.

Author’s Response: Thank you for pointing this out. We have changed this to ‘continuity of care'.

  1. I still find missing a couple of sentences describing the healthcare system in India. Do people have to only pay copayments or are there other types of out-of-pocket expenditure?

Author’s Response: We have now added a description of the health system in India and its payment structure in the Introduction section.

  1. I think listing health expenditure under the independent variables section is strange as it seems to imply that you're predicting multimorbidity based on it.  I think it belongs to the outcomes of interest section. 

Author’s Response: We have now listed healthcare expenditure under the outcome variables section.

  1. Line 165: a space is missing between "variables" and "adjusted".

Author’s Response: We have now added a space. 

  1. The R-squared of the final model seems to be missing. 

Author’s Response: We have now mentioned the value of r2 in the manuscript.

  1. Line 284 is incomplete.

Author’s Response: Thank you for observing this. We have now completed this line.

  1. Line 289"higher healthcare expenditure on multimorbidity" is a bit hard to understand. Higher than what?

Author’s Response: We have now explained this by stating higher than those who did not have multimorbidity.

  1. "A study conducted in the USA revealed a significant decrease in catastrophic health expenditure on multimorbidity among households between 2008 and 2018" It seems like the reason for the decline is unclear. What that due to the ACA? Also, do the authors think that the Indian and USA healthcare systems are comparable?

Author’s Response: Thank you for observing this. We have now removed this sentence and reference as USA and India are not comparable.

  1. The skewness of healthcare expenditure data is a known property and I wouldn't state it as a particular limitation of your study. I think, instead, you may want to highlight that no modeling of healthcare expenditure variable was done, to understand, for example, how family multimorbidity impacts it. And how further research may want to look at that.

Author’s Response: Thank you for your suggestion. We have changed accordingly.

We thank the reviewer for his valuable suggestions and time.